

# Food selectivity and processing by the cold-water coral *Lophelia pertusa*

Dick van Oevelen[1], Christina E. Mueller[1], Tomas Lundälv[2], Jack J. Middelburg[3]

[1]Department of Estuarine and Delta Systems, Royal Netherlands Institute for Sea Research (NIOZ-Yerseke), and Utrecht
University, PO Box 140, 4400 AC Yerseke, The Netherlands
[2]Sven Lovén Centre for Marine Sciences, Tjärnö, University of Gothenburg, 452 96 Strömstad, Sweden
[3]Utrecht University, Department of Earth Sciences, P.O. Box 80.021, 3508 TA Utrecht, The Netherlands

*Correspondence to*: Dick van Oevelen (Dick.van.Oevelen@nioz.nl)

**Abstract.** Cold-water corals form prominent reef ecosystems along ocean margins that depend on suspended resources
produced in surface waters. In this study we investigated food processing of $^{13}C$ and $^{15}N$ labelled bacteria and algae by the
cold-water coral *Lophelia pertusa*. Coral respiration, tissue incorporation of C and N and metabolic-derived C incorporation
into the skeleton were traced following the additions of different food concentrations (100, 300, 1300 µg C l$^{-1}$) and two ratios
of suspended bacterial and algal biomass (1:1, 3:1). Respiration and tissue incorporation by *L. pertusa* increased markedly
following exposure to higher food concentrations. The net growth efficiency of *L. pertusa* was low (0.08±0.03), which is
consistent with their slow growth rates. The contribution of algae and bacteria to total coral assimilation was proportional to
the food mixture in the two lowest food concentrations, but algae were preferred over bacteria as food source at the highest
food concentration. We argue that behavioural responses for these small-sized food particles, such as tentacle behaviour and
mucus trapping, are more likely to explain the observed food selectivity as compared to physical-mechanical considerations.
A comparison of the experimental food conditions to natural organic carbon concentrations above CWC reefs suggests that
*L. pertusa* is well adapted to exploit temporal pulses of high organic matter concentrations in the bottom water caused by
internal waves and downwelling events.

## 1 Introduction

Cold-water corals have a global distribution in the deep sea and are typically found at locations with high bottom-water
velocities such as continental margins, seamounts and mid-ocean ridges (Roberts et al., 2009; Davies and Guinotte, 2011;
Yesson et al., 2012). Some cold-water corals are scleractinians and produce a three-dimensional carbonate structure, which
provides settlement, refuge and feeding ground for many associated organisms (Henry and Roberts, 2007; Kutti et al., 2015).
As a result, these reef communities are diverse, have high biomass and consume up to 20 times more organic carbon per





square meter as compared to surrounding soft-sediment communities (Van Oevelen et al., 2009; White et al., 2012; Cathalot et al., 2015; Rovelli et al., 2015).

The main reef-building coral species in the North Atlantic Ocean is the branching coral *Lophelia pertusa*, which is a passive suspension feeder that uses tentacles to 'catch' particles from the water column. Field observations on stable isotopes and fatty acids suggest that *L. pertusa* feeds on a broad range of food sources including particulate suspended matter, bacteria, phytoplankton and zooplankton (Kiriakoulakis et al., 2005; Duineveld et al., 2007; Sherwood et al., 2008; Dodds et al., 2009). Laboratory studies have confirmed the uptake of suspended particles, bacteria, phytoplankton and zooplankton by cold-water corals (Purser et al., 2010; Mueller et al., 2014; Orejas et al., 2016). Recently, *L. pertusa* was also shown to take up dissolved organic matter in the form of free amino acids (Gori et al., 2014; Mueller et al., 2014) and to fix inorganic carbon into its biomass, supposedly through chemo-autotrophic activity of associated microbes (Middelburg et al., 2015). This flexibility in resource utilization clearly indicates an opportunistic feeding strategy (Mueller et al., 2014; Orejas et al., 2016).

In natural reefs the diversity of organic matter sources is high (Jensen et al., 2012) and it is presently unclear whether cold-water corals exhibit selective resource utilization or feed proportionally to resource availability. Moreover, organic matter supply to cold-water reefs is temporally variable due to seasonality in organic matter production in the surface ocean and the dynamic physical environment in which cold-water reefs occur (Duineveld et al., 2007; Davies et al., 2009; Findlay et al., 2013; Hebbeln et al., 2014; Mohn et al., 2014). Freshly hatched *Artemia salina* nauplii, which are often used as food in aquarium studies of scleractinians, were increasingly taken up by the cold-water coral *L. pertusa* with increasing concentration in the incubation vessel (Purser et al., 2010), indicating that *L. pertusa* responds to changes in food supply. However, to advance our understanding of cold-water coral physiology, we also have to further our knowledge on carbon partitioning within the energy budget of the organism. For the cold-water coral *Desmophyllum dianthus* it was shown that zooplankton contributed to various components of the energy budget, including calcification, respiration and mucus release, following food withdrawal for 1 week (Naumann et al., 2011). The slow (i.e. months) response time of *L. pertusa* to changing food conditions renders this approach less feasible to directly link food uptake to physiological processing (Larsson et al., 2013).

To elucidate the energy budget of *L. pertusa*, we investigated food uptake, food selectivity and subsequent processing with a novel dual isotope labelling technique using mixed diets of $^{13}$C-labeled algae/ $^{15}$N-labeled bacteria and $^{15}$N-labeled algae/$^{13}$C-labeled bacteria. This approach provided the high sensitivity needed to eliminate long-term incubations and allowed us to trace not only uptake but also subsequent processing of algal and bacterial carbon and nitrogen. This experimental mixed diet study better represents the diversity of food available under natural coral reef conditions than traditional single food source studies, yet enables tracing the processing of individual food sources.



## 2 Materials and methods

### 1.1 Experimental design

Our dual isotope tracer design involved exposing separate coral fragments either to a food mixture of $^{13}$C-labelled algae ($^{13}$C-Algae) + $^{15}$N-labelled bacteria ($^{15}$N-Bacteria) or to a mixture of $^{15}$N-labelled algae ($^{15}$N-Algae) + $^{13}$C-labelled bacteria ($^{13}$C-Bacteria) (Fig. 1). Uptake, respiration and calcification rates are subsequently summed to obtain total C or N uptake/processing (i.e. by dividing rates with the fractional $^{13}$C or $^{15}$N enrichment of each food source, see below). Three food concentrations were tested in this study: 8.3 (n=2 per food mixture), 25 (n=3 per food mixture) and 108 (n=3 per food mixture) µmol C l$^{-1}$. The bacterial-C to algal-C ratio was 1:1 in the 8.3 and 25 µmol C l$^{-1}$ treatment, but, due to technical issues, appeared to be 3:1 in the 108 µmol C l$^{-1}$ exposure.

### 1.2 Sampling location and maintenance

Corals were collected at the Tisler Reef, located 70 to 155 m deep in the Skagerrak, at the Norwegian-Swedish border. The Tisler reef is located at a sill, which connects the Kosterfjord deep trough with the open Skagerrak. The current velocity at the reef varies from 0 to 50 cm s$^{-1}$, with peaks in excess of 70 cm s$^{-1}$, and the bottom-water temperature varies normally between 6 to 9 ºC throughout the year (Lavaleye et al., 2009; Wagner et al., 2011) though peaks in excess of 12°C have been observed in recent years (Guihen et al., 2012). The particulate organic carbon (POC) concentration at the reef varies between 3.6 and 8.9 µmol C l$^{-1}$ and the depositional POC fluxes averages at 38 µmol C m$^{-2}$ d$^{-1}$ (Wagner et al., 2011).

Fragments of the cold-water coral *Lophelia pertusa* were collected from a depth of around 110 m using the remotely operated vehicle Sperre Subfighter 7500 DC. Fragments were placed in cooling boxes filled beforehand with cold seawater (7 - 8 ºC) and transported within a few hours to the laboratory at the Sven Lovén Centre in Tjärnö (Sweden). After arrival, the coral fragments were clipped to approximately the same size (7.90±2.12 g dry weight (DW) fragment$^{-1}$; 14.1±2.4 polyps fragment$^{-1}$ as mean ± standard deviation) and were subsequently acclimated for 6 weeks in aquaria (~20 L) placed in a dark thermo-constant room (7 ºC). Sand-filtered (1-2 mm particle size) bottom water from 45 m depth out of the adjacent Koster fjord (salinity 31) was continuously flushed through the aquaria (~1 L min$^{-1}$). Experience at the station and our earlier experiments showed that the sand-filtered water still contains sufficient organic particles, so that no extra food was provided during the acclimation period (Mueller et al., 2014).

### 1.2 Preparation of isotopically labelled algae and bacteria

$^{15}$N-labeled algae were cultured axenically in F/2 culture medium adjusted after Guillard (Guillard, 1975). The culture medium was prepared by replacing 80% of the NaHCO$_3$ ($^{13}$C-Algae) or 70% of the NaNO$_3$ with its heavy isotope equivalent (Cambridge Isotopes, 99% $^{13}$C-NaHCO$_3$, 99% $^{15}$N-NaNO$_3$). Subsequently, a sterile inoculum of the diatom *Thalassiosira*




*pseudonana* (~5 μm) was added. After a 3-week culture period with a 12-hour light-dark cycle, the culture had reached a cell density of $3 - 4 \cdot 10^6$ cells ml$^{-1}$. The diatoms were concentrated by centrifugation at 450 g and the concentrate was rinsed three times with 0.2 μm filtered seawater to remove residual label and the algal suspension was kept frozen until further use.

Bacteria (±1 μm diameter) were cultured by adding a few ml of natural seawater from the Oosterschelde estuary
(Netherlands) to M63 culture medium adjusted after Miller (1972). In the culture medium, 50% of glucose (3 g l$^{-1}$) or 50% of NH$_4$Cl (1.125 g l$^{-1}$) was replaced by its heavy isotope equivalent (Cambridge Isotopes, 99% $^{13}$C-glucose, 99% $^{15}$N-NH$_4$Cl) to obtain $^{13}$C or $^{15}$N isotopically labeled bacteria. After 3 days of culturing in the dark, bacteria were concentrated by centrifugation (14,500 g), rinsed 3 times with 0.2 μm filtered seawater to remove residual label and the bacterial suspension was stored frozen until the start of the experiment.

Subsamples of the algae (n = 3) and bacteria (n = 3) were measured for $^{13}$C, $^{15}$N, C and N (see below). The algae used in the experiment had a molar C:N ratio of 7.8±0.5, 44 at% $^{13}$C and 65 at% $^{15}$N, while bacteria had a C:N ratio of 4.8±0.2, 58 at% $^{13}$C and 47 at% of $^{15}$N.

**1.3 Experimental procedure**

Prior to the start of the experiment incubation chambers (10 L), placed in a temperature-controlled room at 7 ºC, were
filled with 5-μm filtered bottom water from the nearby Koster fjord. Each coral fragment was inserted into an elastic silicone tube, which was mounted on an acrylic plate to allow easy fixing onto the chamber base and to ensure that the fragments retained an upright position. A continuous level of turbulence and water circulation was maintained during the experiment by a motor-driven paddlewheel in the upper part of the incubation chamber (speed: 2 rpm).

The corals were exposed to the isotopically labelled food for 12 h per day during 10 consecutive days (i.e. the 'feeding
period'). A food suspension dosage of a few millilitres was given at the beginning of each day during the feeding period with the respective concentration and ratio of $^{13}$C bacteria/$^{15}$N algae and $^{13}$C algae/$^{15}$N bacteria (see above and Fig. 1). After a 12-hour exposure to the food dosage, the chambers were flushed with 5-μm filtered Koster fjord (140 ml min$^{-1}$) for 12 h to remove food particles, avoid accumulation of waste products and renew the O$_2$ supply. Corals for background isotope measurements (controls) were incubated in parallel under 'acclimatization' conditions: i.e. only exposed to sand-filtered
seawater.

After the last flushing period on day 10, the incubation chambers were closed for 48 hours to measure the production of $^{13}$C dissolved inorganic carbon ($^{13}$C-DIC) as a proxy for respiration (Moodley et al., 2000). Filtered (GF/F) water samples were taken for DIC analysis before (control) and after the incubation period and stored in a 10 ml headspace vial. Biological activity was stopped by adding 10 μl HgCl$_2$ to the vials. The vials were closed with an aluminium cap fitted with a rubber
septum and stored upside down for further analysis. Calculations based on literature respiration rates (Dodds et al., 2007) and pilot experiments indicated that the expected changes in pH, and oxygen and ammonium concentration during the




incubations are limited, so that no negative affect on coral or sponge physiology was expected. Coral fragments were stored frozen (-20°C) at the end of the incubation for further analysis.

### 1.4 Sample analysis

Coral fragments were freeze-dried, weighed and subsequent grinded with a ball Mill for 20 seconds (MM 2000, Retsch, Haan, Germany). This ground coral material, comprised of skeleton and organic tissue, was measured for the incorporation in the skeleton and tissue (following Tanaka et al., 2007 and Mueller et al. 2013). Around 30 mg of a coral sample was transferred to silver measuring boats and measured for C content and $^{13}$C at% using a thermo Electron Flash EA 1112 analyzer (EA) coupled to a Delta V isotope ratio mass spectrometer (IRMS). Another 30 mg of ground coral was transferred to pre-combusted silver boats and gently decalcified by acidification by placing them in an acidic fume for 3 to 4 days to remove most of the inorganic C and then further acidified by stepwise addition of HCl with increasing concentration (maximum concentration 12 mol L$^{-1}$) until the inorganic skeleton was removed (as evidenced by the absence of bubbling after further acid addition) (Mueller et al. 2013). After acidification, the samples were analyzed on the EA-IRMS for C and N content and $^{13}$C and $^{15}$N at% in the organic fraction. Incorporation of $^{13}$C into the inorganic skeleton, as proxy for calcification (sensu Tanaka et al., 2007), was obtained by subtracting the $^{13}$C in the organic fraction from the total $^{13}$C in the ground coral material. Note that this calcification proxy only tracks the incorporation of 'metabolically-derived' carbon, as the $^{13}$C needs to be liberated by metabolism from the organic resource (algae or bacteria) before it can be incorporated. Calcification based on metabolically-derived C may only be a small fraction of total calcification, but it can still be used as a tracer to detect changes in calcification (Mueller et al., 2013).

In the headspace vials taken for DIC analysis, a headspace of ~3 ml was created by injecting N$_2$ gas through the vial septum (Mueller et al., 2013). Samples were acidified with 20 µl of concentrated H$_3$PO$_4$ to transform DIC into gaseous CO$_2$. A 10-µl sample of the headspace was injected into (EA-IRMS) for analysis of CO$_2$ concentration and at% $^{13}$C-CO$_2$.

The incorporation of $^{13}$C and $^{15}$N in coral tissue and $^{13}$C in CaCO$_3$ is the excess (E) $^{13}$C or $^{15}$N in a sample and is calculated as E = F$_{experiment}$ − F$_{background}$, in which F represents the at% of $^{13}$C or $^{15}$N (i.e. $^{13}$C/[$^{12}$C+$^{13}$C] and $^{15}$N/[$^{14}$N+$^{15}$N], respectively) in an experimental or background sample. Hence, E is the above-background at% of $^{13}$C or $^{15}$N and positive values indicate transfer of isotope from the original algal or bacterial source to the coral. The excess values are multiplied with the C or N content in the ground coral material (i.e. µmol C g$^{-1}$ DW and µmol N g$^{-1}$ DW, respectively) and divided with the at% enrichment of the specific food source to obtain total incorporation rate during a 'feeding period' in µmol C g$^{-1}$ DW and µmol N g$^{-1}$ DW. Incorporation rates throughout the paper are expressed as the daily rates by dividing total incorporation with the length of the feeding period (i.e. 10 days). Total respiration is calculated similarly, in which excess values of DIC are multiplied with the DIC concentration (µmol C L$^{-1}$) and chamber volume (10 L). Moreover, daily respiration rates are calculated over the length of the incubation period (2 days) and normalized to the coral (g DW) in an incubation chamber.





**1.5 Data analysis**

Selective uptake of algae or bacteria was assessed with the Chesson index (Chesson, 1983)

$$\alpha_i = \frac{r_i/n_i}{\Sigma_j r_j/n_j} \tag{1}$$

in which $\alpha_i$ is the selectivity index for resource $i$, $r_i$ is the uptake of resource $i$, $n_i$ is the availability of resource $i$ and $j$ is the

total number of resources. The selectivity indices sum to 1 and in the present experiment a selectivity index of 0.5 implies no selectivity, >0.5 indicates 'positive' selectivity (i.e. higher uptake than proportional availability) and <0.5 indicates 'negative' selectivity. This selectivity index normalizes the food uptake against food availability of the different food mixtures for each food concentration tested.

A net growth efficiency (NGE) was calculated from the C incorporation rate into tissue and respiration rates as: NGE =

Tissue incorporation / (Tissue incorporation + Respiration).

Statistical comparisons of processing rates were conducted with an ANOVA in R (R Development Core Team, 2015) using the function *aov* in which food concentration was a continuous factor and with food type as fixed factor. Because the bacteria:algae ratio was not constant over the food concentrations tested (see 'Experimental design' above), we tested processing rates against total food concentration and against bacterial / algal concentration. Data presented are mean ±

standard deviation.

**3 Results**

**3.1 Tissue incorporation and processing of food sources**

Both bacterial and algal C and N were incorporated into the coral tissue and their incorporation rates increased with increasing food concentrations (Fig. 2A, B). Incorporation of algal C increased from 0.013 to 0.09 µg C g$^{-1}$ DW day$^{-1}$ and

bacterial C from 0.017 to 0.14 µg C g$^{-1}$ DW day$^{-1}$ over the food concentration range (Fig. 2A). Carbon incorporation into tissue depended significantly on food concentration (p<0.001, for bacterial C, algal C as well as total C), but not on food type. The incorporation rate into the carbonate skeleton tended to be higher with higher food concentrations, but due to the high variability this increase was not significant (Fig. 2C). Respiration of algal C increased significantly from 0.11 to 0.98 µg C g$^{-1}$ DW day$^{-1}$ and bacterial C from 0.10 to 2.6 µg C g$^{-1}$ DW day$^{-1}$ with increasing food concentration (p<0.001, as well as

for total C concentration, Fig. 2D), but no significant differences were found between food types. A total of 2.6±0.6%, 4.8±0.8% and 3.6±1.4% of the total added organic carbon was recovered in the investigated pools with increasing food concentrations, respectively.

The incorporation rates into tissue were low compared to respiration losses (Fig. 3) resulting in comparatively low net growth efficiencies (NGEs, see Methods for calculation) of 0.08±0.03, independently of food concentration or type (Fig. 3).



### 3.2 Food selectivity

The Chesson index was calculated per food concentration tested and indicates selective uptake as the uptake is normalized to the respective food concentration, i.e. the algae versus bacterial uptake is normalized for the differences in their availability. As such, Chesson values above 0.5 indicating preferential uptake of this food source. We found C-based

Chesson indices ranging from 0.56 to 0.36 for bacteria and 0.44 to 0.64 for algae over the food concentration range (Fig. 4A). A two-factorial ANOVA showed that food concentration did not influence food selectivity, but food type did ($p <$ 0.05), while the interaction term was not significant ($p = 0.09$). When food concentration and food uptake are expressed in N-equivalents, the Chesson index ranged from 0.46 to 0.31 for bacteria and 0.54 to 0.69 for algae (Fig. 4B). A two-factorial ANOVA on food concentration and uptake expressed in N-equivalents, showed that food concentration did not influence

food selectivity, but food type ($p < 0.001$) and the interaction term ($p < 0.01$) were both significant.

### 4 Discussion

#### 4.1 Concentration-dependent food uptake and processing by *Lophelia pertusa*

Higher bacterial and algal concentrations resulted in increased assimilation and respiration rates by *L. pertusa*, indicating that food uptake and metabolism is tightly coupled to food availability. This is consistent with observations by Purser et al.

(2010) and Larsson et al. (2013) showing higher respiration and removal rates of zooplankton with increased particle concentration. Interestingly, food capture rates in Purser et al. (2010) and metabolic activity in this study start to saturate at food concentrations of more than hundred μmol C l$^{-1}$. POC concentrations above CWC reefs vary between 1 to 11 μg C l$^{-1}$ (Kiriakoulakis et al., 2007; Wagner et al., 2011), which implies that *L. pertusa* is well adapted to exploit temporal pulses of high organic matter concentrations in the bottom water caused by internal waves and downwelling event such as observed on

Mingulay reef (Davies et al., 2009), Tisler reef (Wagner et al., 2011) and the Logachev Mounds at Rockall Bank (Soetaert et al.; Duineveld et al., 2007).

In contrast to assimilation and respiration rates, the increase in calcification with food concentration was not significant. Hennige et al. (2014) found a short-term response in respiration rates by *L. pertusa* to higher $p$CO$_2$ conditions, but calcification rates were not significantly affected. Similarly, Larsson et al. (2013) did not find a response of skeletal growth

of *L. pertusa* after long-term exposure (months) to different food concentrations. Hence, it seems that the response time of calcification acts on a longer time scale than does food availability. Also for tropical coral it is known that calcification processes can be less responsive to environmental conditions than tissue growth (Anthony and Fabricius, 2000; Tanaka et al., 2007; Tolosa et al., 2011). One explanation why a longer time period is needed before a response in calcification to altered food conditions can be measured may be the relatively low metabolic costs related to calcification in *L. pertusa* (McCulloch

et al., 2012; Larsson et al., 2013). Naumann et al. (2011) however did measure significantly higher calcification in fed



compared to unfed specimens of the CWC *Desmophyllum dianthus*, but *D. dianthus* is a faster growing species and may therefore respond more rapidly to food availability.

The net growth efficiency (NGE) is the percentage of assimilated organic carbon that is transferred into biomass. Hence, a high NGE means that a food source is efficiently shunted into biomass. We are not aware of NGE estimates for cold-water corals in the existing literature. The NGEs of *L. pertusa* in our study ranged from 4% to 17% and these values are low compared to values of >50% for zooplankton (Anderson et al., 2005), a taxonomic group for which NGE is well studied. The shallow-water anemone *Anthopleura elegantissima*, taxonomically closely related to corals, also has substantially higher NGEs ranging from 30% to 60% (Zamer, 1986). The NGE is positively correlated with growth rate for the sponge *Halichondria panicea* (Thomassen and Riisgard, 1995) and the bivalve *Mytilus edulis* (Kiørboe et al., 1981) and we therefore speculate that the low NGE of the cold-water coral *L. pertusa* is related to their slow growth rates (Roberts et al., 2009). The NGE tended to be higher when *L. pertusa* was feeding on algae compared to bacteria (Fig. 3), but these differences were not significant due to a relatively high variability. Hence, although it is known that the NGE can depend on food quality and quantity (Anderson et al., 2005), additional research is necessary to determine that relation for cold-water corals.

## 4.2 Food-composition dependent uptake by *L. pertusa*

Food assimilation at the lower two food concentrations responded proportionally to the food composition, indicating that *L. pertusa* is an opportunistic and seemingly unselective feeder. This opportunistic feeding strategy is consistent with uptake of various organic resources in aquarium experiments (Gori et al., 2014; Mueller et al., 2014; Orejas et al., 2016) and inferences from natural abundance stable isotope and fatty acid compositions from field-collected CWC (Duineveld et al., 2007; Dodds et al., 2009). Interestingly however, *L. pertusa* fed selectively on algae at higher food concentrations (Fig. 4). This indicates that *L. pertusa* assimilates food in proportion to food availability at comparatively lower food concentrations, but when food is in ample supply, *L. pertusa* starts feeding preferentially on algal organic matter.

Our data do not allow us to identify which mechanisms drive the observed food selectivity. The algal cells (5 μm) are a factor 5 larger in diameter than bacteria, so food size may be a trigger that induces selective behaviour. Consistently, Tsounis et al. (2010) found that several CWCs, amongst other *L. pertusa*, fed at higher rates on adult *Artemia salina* compared to the smaller-sized *A. salina* nauplii. Shimeta and Koehl (1997) conducted a theoretical analysis of selective feeding by passive suspension feeders and found particle selection to be a function of encounter, retention and handling. For the particles considered in this study, i.e. bacteria and algae that are substantially smaller than the feeding tentacles, Shimeta and Koehl predict that encounter rates increase with particle size while particle retention is likely to be independent of particle size. Sole mechanical predictions for the food handling stage provide only part of the story, because behavioural choices may play an important role here (Shimeta and Koehl, 1997). Additional behavioural triggers that may increase encounter and retention





involves enhanced polyp extension in the presence of the preferred food source or environmental conditions (Orejas et al., 2016) and trapping of food particles with the aid of mucus secretion (Mortensen, 2001). Zetsche et al. (2016) recently showed that mucus by *L. pertusa* is produced in small amounts and very localized in response to different stimuli. When exposed to *A. salina* nauplii, mucus strings and so-called 'string balls' were seen to enhance food trapping. Given the comparatively small particle size used in this study, we suggest that behavioural responses are more likely to explain the observed food selectivity as compared to the physical-mechanical considerations.

## 5 Acknowledgements

Lisbeth Jonsson and Ann Larsson are thanked for their assistance with sampling and coral maintenance. Pieter van Rijswijk is thanked for his help whenever it was needed the most. The analytical lab of NIOZ-Yerseke is thanked for sample analysis. This research was supported by the CALMARO project (FP7/2007-2013) within the European Community's Seventh Framework Program (FP7/2007-2013), by the Netherlands Center of Earth System Science and by the Netherlands Organisation for Scientific Research (NWO-VIDI grant no. 864.13.007).

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



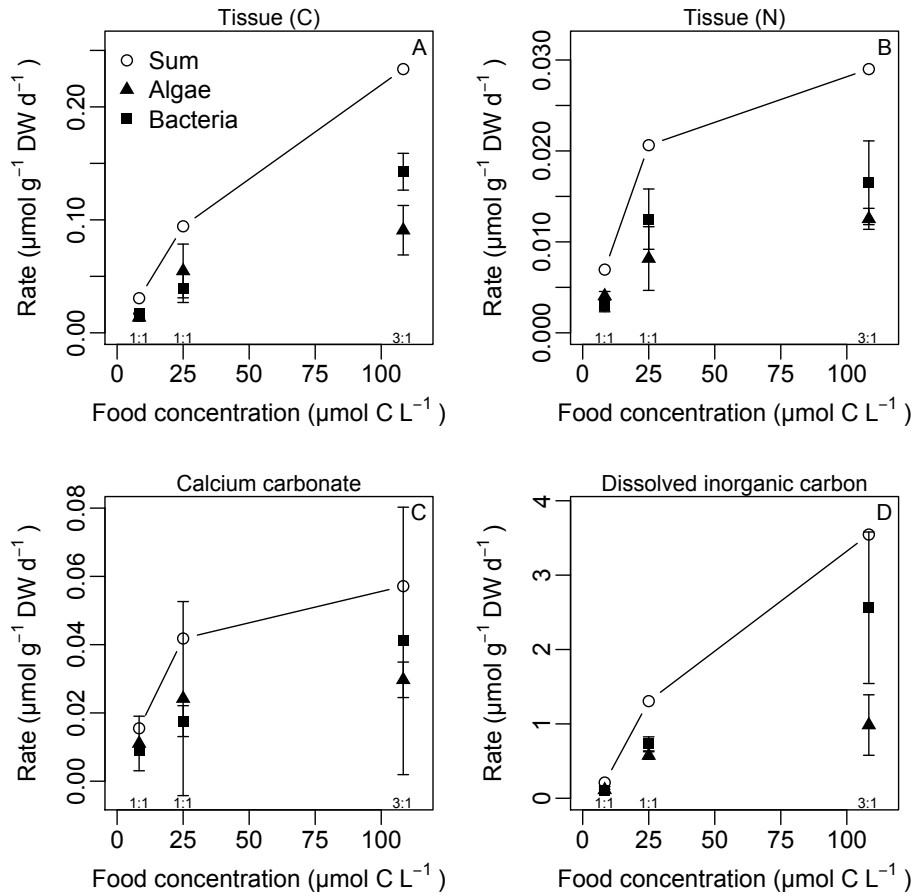

**Figure 2: Food processing by the cold-water coral *L. pertusa* at different food concentrations and compositions. A) Carbon assimilation in tissue, B) nitrogen assimilation in tissue, C) carbon incorporation into coral skeleton, D) carbon respiration. Note that the ratios 1:1 and 3:1 in the subpanels indicate the ratio of bacteria : algae in the respective food concentration treatment.**



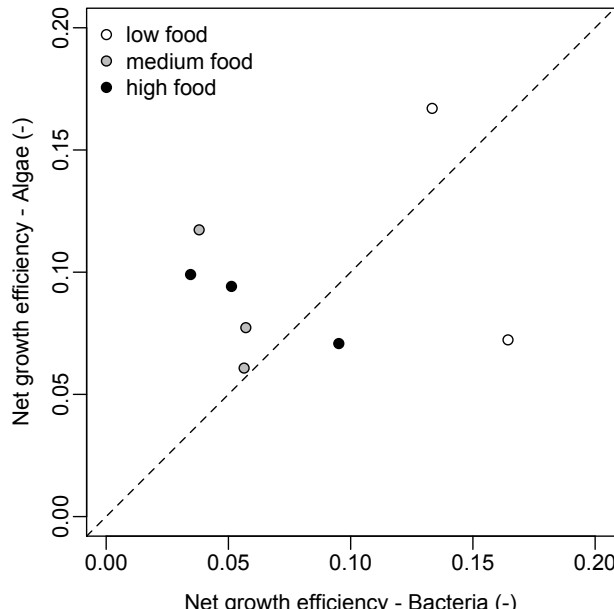

**Figure 3: The net growth efficiency of *L. pertusa* when feeding on algae versus bacteria. The colors represent the low, medium and high food concentration as in Fig. 2.**





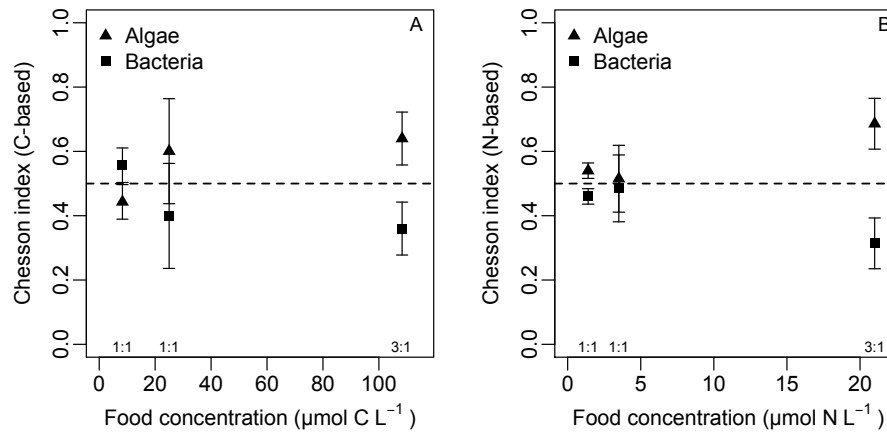

**Figure 4: The Chesson index of *L. pertusa* for feeding on bacteria and algae expressed in A) carbon and B) nitrogen. Note that the ratios 1:1 and 3:1 in both subpanel indicate the ratio of bacteria : algae in the respective food concentration treatment.**