# Peer review of "Food selectivity and processing by the cold-water coral *Lophelia* pertusa"

_Biogeosciences, 2016_

## Referee Comment (RC1) · W.ÂăR. Hunter (Referee) · 25 Aug 2016

**Review of van Oevelen et al. Food Selectivity and processing by the cold-water coral *Lophelia pertusa*.**

**General Comments**

The manuscript by van Oevelen et al. presents a very elegant experiment that tests the food selection and processing by the cold-water coral *Lophelia pertusa*. The experiments conducted provide a novel insight into the whether this coral is capable of feeding selectively and the potential mechanisms that underlie this. The authors' experimental design was excellent, particularly the neat use of $^{13}$C and $^{15}$N tracers to independently trace the uptake of algal and bacterial derived C and N into the corals. One broad concern I have with the paper is the low level of replication within the paper, it would have been nicer to see a greater number of experimental replicates to improve statistical power. I recognise that there are both logistical and ethical considerations to take into account when sampling cold-water corals, but I think the authors need to justify the limited replication within the study. On a further note, I believe the manuscript could be further developed to address how consumer and resource stoichiometry may help to explain the observed differences in food assimilation. It may be useful to look at the relative carbon and nitrogen content of each food source (%C, %N and C:N ratios) and the C:N ratios of the corals, and investigate the changes in food selectivity as responses to these parameters. Overall, I believe this paper is worthy of publication once the authors have addresses the specific comments outlined below.

**Specific Comments**

1. This is an experimental study, as such I believe that the authors need to state a working hypothesis or at the minimum a clearly articulated set of aims. At present the introduction provides a nice review of the current knowledge around *Lophelia pertusa* feeding, but this is not just a descriptive study.

2. I do not believe that the authors' use of analysis of variance is appropriate. I would recommend that the authors remove the statistical tests and seek to describe and interpret the results graphically. Analysis of variance relies upon the assumption that a mean and variance can be estimated from the data.

Given that the lowest food concentrations (8.3 μmol C l$^{-1}$) treatment had only two replicates, this means that estimating a reliable sample mean for this treatment is not possible. Furthermore, given that the bacterial / algal proportions are not repeated over all 3 food concentration treatments, I cannot see how a two-way interaction can be tested within this study. The experimental design is confounded by the fact that the algal:bacterial biomass was 1:1 in two of the treatments but 3:1 in the third. I would ask the authors to acknowledge the limitation this places on the study and interpret their results accordingly.

3. I believe that more could be learned about the feeding responses of L. pertusa by investigating the relative quality of each food source, in terms of average particle size and the %C and %N content. Given that the algal cells where 5 times larger than the bacteria, what can be said about the relative nutritional content of each?

4. Furthermore, some exploration of consumer and resource stoichiometry may help to elucidate selective uptake and incorporation. I would ask the authors to do some data exploration of the C:N ratios and if it is possible to derive a $^{13}$C:$^{15}$N ratio for the food sources and corals. This would potentially allow a greater insight into resource portioning by the corals.

**Minor Comments & Technical Corrections**

Page 1 Line 10: Comma missing – "In this study, we investigated…"

Page 2 Line 14: "it is presently unclear whether cold-water corals exhibit selective resource utilisation or feed proportionally to resource availability" Do you have a reference which would support this supposition.

Page 2 Line 20-21: "However, to advance our understanding of cold-water coral physiology…" This sentence is rather poorly structured. Consider revising to "In order to advance our knowledge of cold-water coral physiology, we must understand how dietary carbon partitioning affects the organismal energy budget." or similar.

Page 2 Line 26-31: Please state the hypothesis for this study.

Page 4 Line 10-13: What were the %C, and %N values of the algal and bacterial food sources?

Page 4 Line 14: Poor grammar, please revise to "Prior to the start of the experiment**,** incubation chambers (10 L) **were** placed in a temperature-controlled room at $7^{\circ}C$ **and**…"

Page 6 Line 2-5: Please can you clarify the terms in the equation. Looking at the I cannot tell if the uptake of resource is per unit time or total? Also is the availability of the resource a ratio or does it have units?

Figure 2: Given that there were only two replicates of the lowest food concentrations (8.3 $\mu$mol C l$^{-1}$) I would suggest that the authors plot the raw data. Sample mean and variance cannot be reliably estimated with less that 3 replicates. This would also apply for figure 4.

---

## Referee Comment (RC2) · E. Gontikaki (Referee) · 7 Sep 2016

**Review of article bg-2016-294 "Food selectivity and processing by the cold-water coral *Lophelia pertusa*"**

Author(s): D. van Oevelen et al.

**General comments**

This is well-written, well-designed concise paper on food selectivity of the CWC *Lophelia pertusa*. The experimental design, experimental procedure and sample and data analysis are sound and reflect the extensive experience of the team of authors on the study of CWC reefs and isotope tracers experiments in general. I highly recommend publication of the present manuscript with only minor modifications.

**Specific comments**

It is mentioned that the selectivity index normalises the food uptake for the differences in the availability of food sources. Does that mean that the different bacterial: algal C ratio in the high load treatment compared to the other two does not affect the result?

**Technical comments**

All the subheadings in section 2 should be corrected (e.g. 2.1 experimental design instead of 1.1).

Page 2, line 24: consider changing the word "feasible" to "effective".

Page 5, line 5: After "incorporation", add "of isotopic tracers".

Page 7, lines 2-4: these lines would fit better into section "2.5 Data Analysis"

Figure 2: For the "sum" (open circles), there are error bars only for the calcium incorporation. Keep it consistent, either present only the mean or add error bars to all graphs. Also, I would remove the lines connecting the "sum" between the treatments, as these are appropriate to use in time series rather than independent concentration treatments.

---

## Author Response (AR1)

Dear Dr. Woulds,

We thank you for your earlier suggestions to clarify our manuscript and the reviewers for their suggestions and favourable evaluation of our manuscript "Food selectivity and processing by the cold-water coral *Lophelia pertusa*". Below, we describe how we will address these issues in the revised version of our manuscript.

To address your initial concern regarding the experimental design, we have added Fig. 1B to clarify the issue. Note that we have also updated Fig. 4 (Fig. 5 in the revision) as a few data points were incorrect in the original figure.

On behalf the authors,
Dick van Oevelen

Reviewer William Hunter

Review of van Oevelen et al. Food Selectivity and processing by the cold-water coral Lophelia pertusa.

General Comments

The manuscript by van Oevelen et al. presents a very elegant experiment that tests the food selection and processing by the cold-water coral *Lophelia pertusa*. The experiments conducted provide a novel insight into the whether this coral is capable of feeding selectively and the potential mechanisms that underlie this. The authors' experimental design was excellent, particularly the neat use of 13C and 15N tracers to independently trace the uptake of algal and bacterial derived C and N into the corals. One broad concern I have with the paper is the low level of replication within the paper, it would have been nicer to see a greater number of experimental replicates to improve statistical power. I recognise that there are both logistical and ethical considerations to take into account when sampling cold-water corals, but I think the authors need to justify the limited replication within the study. On a further note, I believe the manuscript could be further developed to address how consumer and resource stoichiometry may help to explain the observed differences in food assimilation. It may be useful to look at the relative carbon and nitrogen content of each food source (%C, %N and C:N ratios) and the C:N ratios of the corals, and investigate the changes in food selectivity as responses to these parameters. Overall, I believe this paper is worthy of publication once the authors have addresses the
specific comments outlined below.

*We thank the reviewer for his compliments on our experimental work and design. Our experimental design was aimed at a replication at n=3, which we achieved in most cases although we agree that a higher number of replicates would be better. The field station where the experiments were conducted (Tjaerno Marine Laboratory in Sweden) collects corals from the nearby Tisler reef, which is in Norwegian waters. Hence, approval needs to be granted by the Norwegian authorities for the collection of corals. Permission has been given but with restrictions on the amount of corals to collect, so we were unfortunately restricted in the number of replicates we could conduct. We acknowledge this now in the manuscript in the "Materials and methods - Experimental design" by stating* "Replication in this study was limited due to collection restrictions for *Lophelia pertusa* from the Tisler reef." *The stoichiometry remark will be discussed under point 4 below.*

Specific Comments
1. This is an experimental study, as such I believe that the authors need to state a working hypothesis or at the minimum a clearly articulated set of aims. At present the introduction provides a nice review of the current knowledge around Lophelia pertusa feeding, but this is not just a descriptive study.
*We partially agree on this point with the reviewer. It is true that it is not a purely descriptive study, but we do consider this an explorative study to identify feeding preferences and better quantify the energy budget of Lophelia. The reviewer also states, see point 2 below, that we should refrain from statistical testing. Hence, we decided to not end the introduction with a specific hypothesis (that typically requires statistical testing), but better articulated the aims of this study.*

2. I do not believe that the authors' use of analysis of variance is appropriate. I would recommend that the authors remove the statistical tests and seek to describe and interpret the

results graphically. Analysis of variance relies upon the assumption that a mean and variance can be estimated from the data. Given that the lowest food concentrations (8.3 µmol C l-1) treatment had only two replicates, this means that estimating a reliable sample mean for this treatment is not possible. Furthermore, given that the bacterial / algal proportions are not repeated over all 3 food concentration treatments, I cannot see how a two-way interaction can be tested within this study. The experimental design is confounded by the fact that the algal:bacterial biomass was 1:1 in two of the treatments but 3:1 in the third. I would ask the authors to acknowledge the limitation this places on the study and interpret their results accordingly.

*Thanks for pointing this out. We have removed the statistical tests and now take the different food ratios into account when discussing the results in section 4.2, as in section 4.1 we discuss the response to total (i.e. algal + bacterial) food availability.*

3. I believe that more could be learned about the feeding responses of L. pertusa by investigating the relative quality of each food source, in terms of average particle size and the %C and %N content. Given that the algal cells where 5 times larger than the bacteria, what can be said about the relative nutritional content of each?

*Unfortunately, our method of algal and bacteria collection leaves a variable proportion of salt in the residue. Hence, the %C and %N content measurements are unreliable and we therefore decided not to discuss this any further. We however do discuss the effect of cell size on food selectivity in section 4.2. Despite the difference in cell size, mechanical selection cannot explain the observed selectivity.*

4. Furthermore, some exploration of consumer and resource stoichiometry may help to elucidate selective uptake and incorporation. I would ask the authors to do some data exploration of the C:N ratios and if it is possible to derive a 13C:15N ratio for the food sources and corals. This would potentially allow a greater insight into resource portioning by the corals.

*This is a good point and we looked further into this. As a result, we found a surprising uncoupling of C and N uptake in one treatment (see the new figure 4). We explicitly state that the stoichiometric effects are clearly visible, but it does not modifiy the observed food selectivity at higher food concentrations.*

Minor Comments & Technical Corrections
Page 1 Line 10: Comma missing – "In this study, we investigated…" *OK*

Page 2 Line 14: "it is presently unclear whether cold-water corals exhibit selective resource utilisation or feed proportionally to resource availability" Do you have a reference which would support this supposition. *No, we were referring to the fact that there have not been studies that have assessed this. We have modified the sentence accordingly.*

Page 2 Line 20-21: "However, to advance our understanding of cold-water coral physiology…" This sentence is rather poorly structured. Consider revising to "In order to advance our knowledge of cold-water coral physiology, we must understand how dietary carbon partitioning affects the organismal energy budget." or similar. *OK*

Page 2 Line 26-31: Please state the hypothesis for this study. *See our response above.*

Page 4 Line 10-13: What were the %C, and %N values of the algal and bacterial food sources? *See our response above.*

Page 4 Line 14: Poor grammar, please revise to "Prior to the start of the experiment, incubation chambers (10 L) were placed in a temperature-controlled room at 7oC and…" *OK*

Page 6 Line 2-5: Please can you clarify the terms in the equation. Looking at the I cannot tell if the uptake of resource is per unit time or total? Also is the availability of the resource a ratio or does it have units? *We have better explained the terms in this equation.*

Figure 2: Given that there were only two replicates of the lowest food concentrations (8.3 µmol C l-1) I would suggest that the authors plot the raw data. Sample mean and variance cannot be reliably estimated with less that 3 replicates. This would also apply for figure 4. *We tried this, but the figure becomes quite messy and unclear. To address this comment we have therefore noted in the legend explicitly that the mean ± 'sd' is mean ± range for the low food treatment. In addition, we have added Fig.1B to clarify the design.*

Reviewer Evina Gontikaki

General comments

This is well-written, well-designed concise paper on food selectivity of the CWC *Lophelia pertusa*. The experimental design, experimental procedure and sample and data analysis are sound and reflect the extensive experience of the team of authors on the study of CWC reefs and isotope tracers experiments in general. I highly recommend publication of the present manuscript with only minor modifications.

*We thank the reviewer for these compliments on our experimental work and design.*

Specific comments

It is mentioned that the selectivity index normalises the food uptake for the differences in the availability of food sources. Does that mean that the different bacterial: algal C ratio in the high load treatment compared to the other two does not affect the result?
*Correct*

Technical comments

All the subheadings in section 2 should be corrected (e.g. 2.1 experimental design instead of 1.1).
*This has now been corrected.*

Page 2, line 24: consider changing the word "feasible" to "effective".
*Changed accordingly.*

Page 5, line 5: After "incorporation", add "of isotopic tracers".
*Changed accordingly.*

Page 7, lines 2-4: these lines would fit better into section "2.5 Data Analysis"
*Changed accordingly.*

Figure 2: For the "sum" (open circles), there are error bars only for the calcium incorporation. Keep it consistent, either present only the mean or add error bars to all graphs. Also, I would remove the lines connecting the "sum" between the treatments, as these are appropriate to use in time series rather than independent concentration treatments.
*The figures are consistent, i.e. also in the calcium carbonate figure there are only error bars (though large) in the C-algae and C-bacteria data.*
*We disagree that it would be inappropriate to draw lines between the concentration treatments. There is a large body of scientific work on functional responses (e.g. Holling type functional responses or Michaelis-Menten kinetics) that is focussed on relating uptake to concentration.*

[revised manuscript text omitted]

**A)**

**Carbon uptake and processing**

| C-Alg: | C-tissue | C-DIC | C-CaCO₃ |
| C-Bac: | C-tissue | C-DIC | C-CaCO₃ |

**Nitrogen uptake**

| N-Alg: | N-tissue |
| N-Bac: | N-tissue |

**B)**

| Food | ¹³C-Alg + ¹⁵N-Bac | ¹⁵N-Alg + ¹³C-Bac |
|------|-------------------|-------------------|
| Low | | |
| Medium | | |
| High | | |

Figure 1: Experimental design of the dual-isotope labelling study. A) Different coral fragments were exposed to a food mixture of ¹³C-labelled algae (¹³C-Alg) + ¹⁵N-labelled bacteria (¹⁵N-Bac) or to a mixture of ¹⁵N-labelled algae (¹⁵N-Alg) + ¹³C-labelled bacteria (¹³C-Bac). The N uptake in tissue (N-tissue), C uptake in tissue (C-tissue), respiration to dissolved inorganic carbon (C-DIC) and C incorporation into the skeleton (C-CaCO₃) was calculated for each incubation from the ¹³C and ¹⁵N transfer (see Methods). B) Full experimental design with 8 incubations for the ¹³C-Alg + ¹⁵N-Bac treatment and 8 incubations for the ¹⁵N-Alg + ¹³C-Bac treatment, partitioned over a low, medium and high food concentration treatment.

[Figure]

[Figure]

Figure 2: Food processing by the cold-water coral *L. pertusa* at different food concentrations and compositions. A) Carbon assimilation in tissue, B) nitrogen assimilation in tissue, C) carbon incorporation into coral skeleton, D) carbon respiration. The mean ± range is shown for the low food concentration treatment (n = 2). The ratios 1:1 and 3:1 in the subpanels indicate the ratio of bacteria : algae in the respective food concentration treatment.

[Figure]

**Figure 3: The net growth efficiency of *L. pertusa* when feeding on algae versus bacteria. The colors represent the low, medium and high food concentration as in Fig. 2.**

[Figure]

**Figure 4: Resource stoichiometry in the experiment, presented as the C:N ratio of the resource versus the C:N ratio in the coral tissue after the incubation. The grey bar is the C:N ratio of the suspended food (i.e. algae + bacteria), the blue dots are the C:N ratios of the coral tissue in the treatment [13]C-algae + [15]N-bacteria and the red dots are the C:N ratios of the coral tissue in the treatment [15]N-algae + [13]C-bacteria.**

[Figure]

[Figure]

**Figure 5: The Chesson index of *L. pertusa* for feeding on bacteria and algae expressed in A) carbon and B) nitrogen.** The mean ± range is shown for the low food concentration treatment (n = 2). The ratios 1:1 and 3:1 in both subpanel indicate the ratio of bacteria : algae in the respective food concentration treatment.

[Figure]

**Carbon uptake and processing**

$^{13}$C-Alg: | $^{13}$C-tissue | $^{13}$C-DIC | $^{13}$C-CaCO$_3$

$^{13}$C-Bac: | $^{13}$C-tissue | $^{13}$C-DIC | $^{13}$C-CaCO$_3$

**Nitrogen uptake**

$^{15}$N-Alg: | $^{15}$N-tissue

$^{15}$N-Bac: | $^{13}$N-tissue

[Figure]

**Figure 4: The Chesson index of *L*.**